Identification of a novel inhibitor of SARS-CoV-2 3CL-PRO through virtual screening and molecular dynamics simulation

http://orcid.org/0000-0001-5656-1833 Bepari Asim Kumar asim.bepari@northsouth.edu
Reza Hasan Mahmud
Department of Pharmaceutical Sciences, North South University , Dhaka , Bangladesh
Silva Pedro
Electronic publication date: 2021 Apr 13
Publication date: 2021
Volume: 9
Electronic Location ID: e11261
Received 2020 Nov 9; Accepted 2021 Mar 22
Copyright: © 2021 Bepari and Reza
Copyright year: 2021
Copyright holder: Bepari and Reza
License: This is an open access article distributed under the terms of the Creative Commons Attribution License, which permits unrestricted use, distribution, reproduction and adaptation in any medium and for any purpose provided that it is properly attributed. For attribution, the original author(s), title, publication source (PeerJ) and either DOI or URL of the article must be cited.
License URL: https://creativecommons.org/licenses/by/4.0/

Keywords: COVID-19, Main protease, Mpro, docking, Coronavirus, in silico, SARS-CoV-2, 3CL-PRO, Vina, Gromacs

Funding: The authors received no funding for this work.

==============================
Background

The COVID-19 pandemic, caused by the SARS-CoV-2 virus, has ravaged lives across the globe since December 2019, and new cases are still on the rise. Peoples’ ongoing sufferings trigger scientists to develop safe and effective remedies to treat this deadly viral disease. While repurposing the existing FDA-approved drugs remains in the front line, exploring drug candidates from synthetic and natural compounds is also a viable alternative. This study employed a comprehensive computational approach to screen inhibitors for SARS-CoV-2 3CL-PRO (also known as the main protease), a prime molecular target to treat coronavirus diseases.

Methods

We performed 100 ns GROMACS molecular dynamics simulations of three high-resolution X-ray crystallographic structures of 3CL-PRO. We extracted frames at 10 ns intervals to mimic conformational diversities of the target protein in biological environments. We then used AutoDock Vina molecular docking to virtual screen the Sigma–Aldrich MyriaScreen Diversity Library II, a rich collection of 10,000 druglike small molecules with diverse chemotypes. Subsequently, we adopted in silico computation of physicochemical properties, pharmacokinetic parameters, and toxicity profiles. Finally, we analyzed hydrogen bonding and other protein-ligand interactions for the short-listed compounds.

Results

Over the 100 ns molecular dynamics simulations of 3CL-PRO’s crystal structures, 6LZE, 6M0K, and 6YB7, showed overall integrity with mean Cα root-mean-square deviation (RMSD) of 1.96 (±0.35) Å, 1.98 (±0.21) Å, and 1.94 (±0.25) Å, respectively. Average root-mean-square fluctuation (RMSF) values were 1.21 ± 0.79 (6LZE), 1.12 ± 0.72 (6M0K), and 1.11 ± 0.60 (6YB7). After two phases of AutoDock Vina virtual screening of the MyriaScreen Diversity Library II, we prepared a list of the top 20 ligands. We selected four promising leads considering predicted oral bioavailability, druglikeness, and toxicity profiles. These compounds also demonstrated favorable protein-ligand interactions. We then employed 50-ns molecular dynamics simulations for the four selected molecules and the reference ligand 11a in the crystallographic structure 6LZE. Analysis of RMSF, RMSD, and hydrogen bonding along the simulation trajectories indicated that S51765 would form a more stable protein-ligand complexe with 3CL-PRO compared to other molecules. Insights into short-range Coulombic and Lennard-Jones potentials also revealed favorable binding of S51765 with 3CL-PRO.

Conclusion

We identified a potential lead for antiviral drug discovery against the SARS-CoV-2 main protease. Our results will aid global efforts to find safe and effective remedies for COVID-19.

Introduction

The “severe acute respiratory syndrome coronavirus 2” (SARS-CoV-2), responsible for the coronavirus disease-2019 (COVID-19), originated in Wuhan, China in late 2019 as a pneumonia outbreak causing acute respiratory distress syndrome and related complications (Huang et al., 2020; Zhou et al., 2020; Wu et al., 2020; Gorbalenya et al., 2020). Considering the severity of symptoms among the affected people and rapid spread, the World Health Organization (WHO) declared COVID-19 as a pandemic on 11 March 2020. This catastrophe has created an unprecedented healthcare crisis confounded with multifaceted economic, social, and cultural impacts (Sultana & Mahmud Reza, 2020; McKibbin & Fernando, 2020; Hartley & Perencevich, 2020; Headey et al., 2020; Forster et al., 2020). Despite extensive measures taken at individual to global scales, the world has only a few arsenals to fight against this massive disaster. While remdesivir, the only FDA-approved drug to treat COVID-19, is indicated for patients 12 years of age and older requiring hospitalization, we all are in pursuit of safer and more effective antiviral agents.

SARS-CoV-2 virus is closely related to other coronaviruses, including SARS-CoV and MERS-CoV, and carries a single-stranded RNA genome of ∼30 kb, which encodes at least 14 open-reading frames (ORFs) (Zhou et al., 2020; Wu et al., 2020; Kim et al., 2020; Gordon et al., 2020). ORF1a and ORF1ab produce polypeptides pp1a and pp1ab, respectively, which generate nonstructural proteins (nsps) upon proteolytic cleavage and form the replicase–transcriptase complex (Kim et al., 2020; Gordon et al., 2020; Jiang et al., 2020). The activity of 3CL-PRO (also known as 3C-like proteinase, main protease, and Mpro) is crucial in the auto-proteolysis of viral polypeptides and is a prime target in the discovery of antiviral agents for COVID-19 (Ziebuhr, Snijder & Gorbalenya, 2000; Anand et al., 2003; Zhang et al., 2020; Jin et al., 2020).

Many high-resolution X-ray crystallographic structures of SARS-CoV-2 3CL-PRO, in both bound and unbound states, are available in the Protein Data Bank (PDB) (www.wwpdb.org). These three-dimensional structures can significantly help design, discover, and develop potential inhibitors for future therapeutic applications. Computational methods are introducing many quick and efficient avenues to reach destinations in the journey of drug discovery and development (Kapetanovic, 2008; Macalino et al., 2015; Yu & MacKerell, 2017; Cui et al., 2020). It is noteworthy that proteins are dynamic in a biological environment, in contrast to the static X-ray crystallographic structures. Virtual screening methods for approved drugs or large databases such as ZINC15 usually involve only a few target structures; therefore, they are more likely to leave off potential ligands. In this study, we have employed a comprehensive in silico approach to identify leads for the treatment of COVID-19 through inhibition of the viral main protease. We generated multiple target structures through molecular dynamics simulations of 3CL-PRO crystal structures and performed target-based virtual screening of the MyriaScreen Diversity Library II. Top compounds were then scrutinized for physicochemical properties, pharmacokinetic profiles, and toxicity risks. Subsequently, we performed protein-ligand interaction analyses for the best picks. Results from this comprehensive computational analysis may assist in finding an effective therapeutic intervention for COVID-19.

Materials and Methods

Protein structure

We retrieved X-ray crystallographic protein structures with PDB IDs 6LZE (Dai et al., 2020), 6M0K (Dai et al., 2020), and 6YB7 from the Protein Data Bank (www.rcsb.org). A multiple structure alignment was done using the mTm-align webserver (Dong et al., 2018a).

Ligand libraries

MyriaScreen Diversity Library II is a powerful resource for lead discovery (Screening Compounds, 2020). Upon request to Sigma-Aldrich, we received an sdf file of this library which contains 10,000 high-purity screening compounds. Sigma–Aldrich constructed this popular library from over 300,000 compounds on the basis of diversity and drug-likeness. All structures were edited using Open babel (O’Boyle et al., 2011) and Discovery Studio Visualizer (Discovery Studio Visualizer, v20.1.0.192, 2019; BIOVIA, Dassault Systèmes, San Diego, CA, USA).

Virtual screening

All non-amino acid residues from a protein structure were removed using UCSF Chimera alpha version 1.14 (2019) (Pettersen et al., 2004). Then the Dock Prep tool of the Chimera program was used to prepare the protein for docking. All default parameters were selected and the structure was saved as a pdb file. In AutoDockTools version 1.5.6 (Morris et al., 2009) the pdb file was then edited by adding polar hydrogens, merging non-polar hydrogens and adding Kollman charges. The final macromolecule was saved in the pdbqt format.

We used Parallelized Openbabael and Autodock suite Pipeline (POAP) to automate the AutoDock Vina virtual screening process (Samdani & Vetrivel, 2018). The Ligand Preparation Module of POAP prepared the ligands by adding hydrogens, generating 3D coordinates and minimizing energy. Ligand files were saved in the pdbqt format. Then we used the Virtual screening Module of POAP to screen the ligands using AutoDock Vina (Trott & Olson, 2010). The inhibitor 11a complexed with 6LZE was used as a guide to make the grid box. For the grid box, the spacing was set at default 1 Å, center xyz coordinates were 10.700, 0.784, 23.667, and the dimension was 26 × 26 × 26. Exhaustiveness was set at eight. Ligands were ranked based on the binding energy (kcal/mol). A more negative value indicates stronger protein-ligand binding.

We performed rigid docking for the best four ligands and the reference inhibitor 11a using AutoDock4.2 (Morris et al., 2009). We used the same ligand and protein files prepared for the Vina virtual screening. For the grid parameter file (.gpf), atom types were selected from the ligands files, the grid was centered on the ligand, grid dimension was 60 × 60 × 60, and the spacing was 0.375 Å. The Lamarckian Genetic Algorithm (LGA) was used for the simulation and the maximum number of energy evaluations was 2,500,000. The best docked poses were selected based on the binding scores and complexes were generated. Subsequently, we used those complexes for protein-ligand interaction analyses in Discovery Studio Visualizer.

Molecular dynamics simulations

Molecular dynamics (MD) simulations were carried out using GROMACS (Berendsen, Van der Spoel & Van Drunen, 1995; Abraham et al., 2015; Lindahl & Van der Spoel, 2019) and a high-performance computing system equipped with an Intel Xeon CPU and an NVIDIA Tesla K40c GPU. For the best four ligands, we used the protein-ligand complexes generated in AutoDock4.2 docking. For the reference complex 6LZE-11, we used the PDB structure. Protein topologies were prepared by the pdb2gmx module of GROMACS using the CHARMM36 all-atom force field (Vanommeslaeghe et al., 2010) and the TIP 3-point water model. Ligand topologies were generated by the CHARMM General Force Field (CGenFF) program version 2.4.0 (“CGenFF Home”, https://cgenff.umaryland.edu/). A dodecahedron box was defined where the protein was positioned at least 1.0 nm from the box edge, filled with approximately 20,000 water molecules, and four sodium ions were added to neutralize the overall charge. The simulation system was energy minimized with a maximum 50,000 steps of steepest descent minimization algorithm. The solvent and ions were equilibrated in two restrained phases. The reference temperature was 300 K for the NVT (isothermal-isochoric) ensemble and the reference pressure was 1.0 bar for the subsequent NPT (isothermal-isobaric) ensemble. Finally, we performed unrestrained MD simulations of the equilibrated systems. Leap-frog integrator was used with a step size of 2 fs. Constraint algorithm was LINCS for NVT, NPT, and the production MD runs. The short-range van der Waals cutoff was 1.2 nm. Modified Berendsen thermostat was used for temperature coupling and Parrinello–Rahman barostat was used for pressure coupling. Similar MD parameters were also used in other studies (Selvaraj et al., 2020; Joshi et al., 2020).

We analyzed simulation trajectories using the GROMACS analysis tools. We also used VMD (Humphrey, Dalke & Schulten, 1996) for analyzing protein-ligand hydrogen bonding.

MMPBSA binding energy calculation

Binding free energy for protein-ligand complexes was computed using the g_mmpbsa tool (Kumari, Kumar & Lynn, 2014). We calculated free energy from MD trajectories separately on two periods, 20–25 ns and 45–50 ns, by sampling snapshots at every 100 ps. The binding free energy was calculated as the sum of van der Waal energy, electrostatic energy, polar solvation energy, and the solvent accessible surface area (SASA) energy.

Results

Molecular dynamics simulations of SARS-CoV-2 3CL-PRO

To predict the dynamics and stability of SARS-CoV-2 3CL-PRO, we performed GROMACS molecular dynamics (MD) simulations of three high-resolution structures with PDB IDs 6LZE (1.5 Å), 6M0K (1.5 Å), and 6YB7 (1.25 Å). 6YB7 represents an apo form with unliganded active sites, whereas 6LZE and 6M0K are holo forms complexed with inhibitors 11a and 11b, respectively. Visualization and alignment indicated significant agreement among the structures (Fig. 1A) with an average pairwise RMSD of 0.52 angstroms and a TM-score of 0.985 (on a scale of 0–1). A protein chain was isolated from the complex, and the topology was prepared using the CHARMM-36 force field. The protein was solvated in a water box with appropriate ions to simulate the biological system. The system’s potential energy converged very quickly, within 1,000 steps (Fig. 1B), to relax the protein-water system by eliminating unusual steric clashes. During NVT (constant number of particles, volume, and temperature) equilibration, the temperature reached 300 K before 10 ps and was maintained (Fig. 1C). Subsequently, the system underwent an equilibration at an NPT (isothermal-isobaric) ensemble, where the system pressure plateaued at 1 bar with some fluctuations (Fig. 1D). These results indicated that the simulation system was well prepared, albeit some minor variations, for the selected protein structures.

Figure 1 Molecular dynamics simulation of 3CL-PRO’s three crystal structures (6LZE, 6M0K, and 6YB7).

(A) Alignment of three crystal structures. (B) Energy minimization for molecular dynamics simulation. (C) NVT equilibration. (D) NPT equilibration. (E–J) Conformational changes of four amino acid residues at the active site of 3CL-PRO over the simulation period. (K) RMSD (running averages) of alpha carbons. (L) RMSF of alpha carbons. Inset shows fluctuations of a loop region of 6LZE.

Next, we proceeded with the 100 ns production MD simulations and the output trajectories were analyzed for various features of the simulation. Visual inspection of frames extracted at different time intervals provides an idea of the dynamics the protein is undergoing in a biological system. For instance, Figs. 1E–1J show orientations of the residues GLY143, CYS145, HIS164, and GLU166, which play critical roles in inhibitor binding, at 20 ns intervals. Changes were apparent for GLU166, compared to other labeled residues. Presumably, conformational alterations are apparent for loop regions. We calculated the root-mean-square deviation (RMSD) of all Cα atoms in the trajectory in reference to the alpha carbons of energy minimized proteins (Fig. 1K). Average RMSD values for 6LZE, 6M0K, and 6YB7 were 1.96 (± 0.35) Å, 1.98 (± 0.21) Å, and 1.94 (± 0.25) Å, respectively, indicating overall stability. We also calculated the root-mean-square fluctuation (RMSF), a measure of standard deviations of atomic positions in the trajectory from the reference frames, for the Cα domains (Fig. 1L). RMSF values rarely crossed 2 Å for most of the atoms. Mean (± SD) RMSF values were 1.21 ± 0.79 (6LZE), 1.12 ± 0.72 (6M0K), and 1.11 ± 0.60 (6YB7). We observed very high fluctuations at extreme ends, which is a usual phenomenon. For 6LZE, there is also a spike for atom numbers 567–797, corresponding to the residues from 44 to 53. Again, this is an expected behavior for a protein’s loop regions (Fig. 1L, inset).

Together, our results from molecular dynamics simulations infer integrity of SARS-CoV-2 3CL-PRO crystal structures. Nevertheless, the conformations showed some alterations over the 100 ns simulation period, which could have significant biological implications in protein-ligand interactions.

AutoDock vina virtual screening of the myriascreen diversity library II

MyriaScreen diversity library II comprises 10,000 high-purity compounds suitable for lead discovery. In the first phase, we screened the whole library with the virtual screening module of POAP using AutoDock Vina against three crystal structures, 6LZE, 6M0K, and 6YB7, of 3CL-PRO. Filtering with an average predicted binding affinity of −8 kcal/mol or lower generated a combined top list of 286 ligands.

We extracted frames at 10 ns intervals from the MD simulation trajectories of 6LZE, 6M0K, and 6YB7, which yielded 30 pdb files for the second screening phase. Plus, we included three crystal structures and performed Vina molecular docking to virtual screen the top 286 compounds against 33 target structures for 3CL-PRO. The top 20 compounds based on overall binding affinities are listed in Table 1. The first molecule, R897698, and the last molecule, R461083, showed binding affinities of −8.7 and −8.0 kcal/mol, respectively. We observe considerable deviations in binding affinities among the 33 protein structures for individual small molecules. Overall, top 20 ligands showed greater affinities to 6LZE compared to other structures (Table 1).

Table 1 Top ligands from the virtual screening of MyriaScreen Diversity Library II against 33 structures of 3CL-PRO.

		Predicted binding affinity (kcal/mol)	
		Total	Single crystal structures	MD simulation structures (Average of ten structures)	
Rank	Ligand ID	Average	SD	6LZE	6M0K	6YB7	6LZE_MD	6M0K_MD	6BY7_MD	
1	R897698	−8.7	0.5	−9.7	−9.0	−8.3	−8.8	−8.3	−8.9	
2	ST031238	−8.4	0.5	−9.2	−8.7	−8.2	−8.6	−8.3	−8.4	
3	ST042014	−8.4	0.6	−9.8	−9.3	−8.2	−8.6	−7.9	−8.3	
4	ST018363	−8.3	0.7	−9.7	−8.8	−8.5	−8.5	−7.6	−8.3	
5	L363340	−8.2	0.6	−9.2	−8.4	−7.9	−8.2	−7.9	−8.3	
6	ST031351	−8.1	0.5	−8.7	−7.9	−9.3	−8.3	−7.6	−8.2	
7	L220477	−8.1	0.8	−9.4	−8.7	−8.0	−8.4	−7.2	−8.4	
8	R679445	−8.1	0.6	−8.8	−8.5	−8.5	−8.3	−7.5	−8.3	
9	ST000954	−8.1	0.6	−9.0	−8.4	−8.8	−8.4	−7.6	−8.0	
10	R872172	−8.1	0.6	−9.2	−8.9	−8.4	−8.2	−7.5	−8.3	
11	ST074801	−8.1	0.5	−8.9	−8.0	−8.1	−8.3	−7.6	−8.3	
12	ST088323	−8.1	0.5	−8.9	−8.6	−8.4	−8.1	−7.6	−8.2	
13	ST018407	−8.1	0.6	−8.8	−8.7	−8.8	−8.2	−7.9	−7.9	
14	S51765	−8.0	0.6	−8.6	−8.2	−8.6	−8.2	−7.5	−8.1	
15	ST074799	−8.0	0.5	−8.8	−8.1	−8.9	−8.1	−7.6	−8.2	
16	R818984	−8.0	0.7	−9.2	−9.1	−9.0	−8.1	−7.4	−8.2	
17	ST094780	−8.0	0.6	−8.6	−8.2	−8.3	−8.3	−7.3	−8.1	
18	ST020475	−8.0	0.6	−9.3	−9.1	−8.3	−8.1	−7.4	−8.1	
19	L128643	−8.0	0.7	−9.9	−8.4	−8.5	−8.0	−7.6	−7.9	
20	R461083	−8.0	0.5	−9.1	−8.4	−8.3	−8.1	−7.5	−8.0	

The predictive performance in virtual screening can vary greatly depending on many factors including the target structure, the docking tool, and the docking protocol. We used AutoDock Vina, a free, open source, widely cited, and one of the most efficient docking tools (Durrant et al., 2013; Wang et al., 2016). To validate the screening protocol, we separated the co-crystallized ligand 11a from the PDB structure 6LZE and then re-docked using the same protocol employed for the virtual screening. We superimposed the docked complex to the crystal structure in Pymol. Indeed, the binding pose generated by Vina was a close match with the crystal structure (Fig. S1).

For further validation of the Vina virtual screening protocol, we retrieved 50 decoys from the DUD-E database (Mysinger et al., 2012) by supplying 11a as the active ligand. We compared binding scores of 11a, the top 20 hits from the two phase of Vina screening, and 50 decoys (Table S1). Interestingly, when we considered all 33 protein structures, all of the top 20 ligands had superior average scores than the decoys. The reference molecule scored better than most of the decoys, although, nine of the 50 decoys topped 11a by slight margins. To the contrary, 19 decoys scored higher than 11a when we considered only the 6LZE crystal structure. Therefore, our virtual screening protocol seemed to produce reasonable predictive power for the selected ligands and target structures of 3CL-PRO.

In silico ADME/Tox profiling

We predicted pharmacokinetic parameters of small molecules through the SwissADME webserver (Daina, Michielin & Zoete, 2017). Table 2 shows physicochemical and solubility descriptors for the top 20 ligands. Molecular weight varied between 367 and 525 g/mol, which falls within the optimum range (200–600 g/mol) for druglikeness. The number of rotatable bonds indicates a structure’s flexibility, and compounds with 10 or fewer rotatable bonds are considered candidates for good oral bioavailability in rats . Khanna & Ranganathan (2009) showed that the mean number of rotatable bonds was seven for drugs and three for toxins (Khanna & Ranganathan, 2009). We found the number of rotatable bonds in the range of 0–7 for the top ligands (Table 2). The numbers of H-bond acceptors and donors were 3–9 and 0–4, respectively. The topological polar surface area (TPSA) values are based on the polar fragments’ surface contributions and indicate the overall polarity of a compound. Table 2 demonstrates that TPSA values were relatively higher for most of the top-ranked ligands, highest for #2 and lowest for #19. Lipophilicity, usually expressed as LogPo/w, is a crucial determinant of a drug’s pharmacokinetic and pharmacodynamic profiles. There are different methods for the prediction of LogP. Table 2 displays WLOGP (Wildman & Crippen, 1999) and consensus LogPo/w of our virtual screening’s top compounds. A drug’s solubility is better when LogP is less than three, whereas a LogP in the range of −1 to 5.9 enhances membrane permeability (Arnott & Planey, 2012). All compounds in our list conform to the requirements for lipid solubility. Table 2 also demonstrates ESOL LogS values and solubility categories for the top list. The minimum and maximum LogS values were −7.57 and −3.77 for ligand #16 and #9, respectively. ESOL estimates the aqueous solubility of a lead directly from the chemical structure (Delaney, 2004). Thus, ligand #9 is the most water soluble compound among the hits from our virtual screening.

Table 2 Computed physicochemical properties of top ligands.

		Physicochemical properties	Lipid solubility	Water solubility	
Rank	Ligand ID	#Rotatable bonds	#H-bond acceptors	#H-bond donors	TPSA	WLOGP	Consensus LogP	ESOL LogS	ESOL Class	
1	R897698	4	6	0	120.6	5.31	3.31	−6.41	Poorly soluble	
2	ST031238	5	7	1	149.68	1.57	1.45	−4.26	Moderately soluble	
3	ST042014	7	9	2	134.17	4.61	2.82	−4.97	Moderately soluble	
4	ST018363	5	8	1	108.37	6.54	4.11	−6.46	Poorly soluble	
5	L363340	4	5	0	101.2	5.88	4.92	−6.5	Poorly soluble	
6	ST031351	5	7	1	137.57	3.25	2.43	−4.81	Moderately soluble	
7	L220477	3	5	1	112.37	4.6	3.79	−5.94	Moderately soluble	
8	R679445	3	4	1	140.81	5.69	5.75	−7.1	Poorly soluble	
9	ST000954	5	6	4	141.12	−1.51	0.79	−3.77	Soluble	
10	R872172	5	3	0	62.34	5.94	5.14	−7.08	Poorly soluble	
11	ST074801	6	4	1	95.2	2.99	3.09	−4.57	Moderately soluble	
12	ST088323	4	8	2	137.57	1.75	2.02	−3.9	Soluble	
13	ST018407	5	4	1	68.27	7.39	6.11	−7.36	Poorly soluble	
14	S51765	0	7	0	115.56	4.66	3.66	−4.91	Moderately soluble	
15	ST074799	6	3	1	71.41	3.73	4.02	−5.23	Moderately soluble	
16	R818984	4	3	0	51.44	6.1	5.29	−7.57	Poorly soluble	
17	ST094780	3	5	1	69.72	3.04	3.59	−5.23	Moderately soluble	
18	ST020475	4	5	2	99.85	2.89	2.98	−4.9	Moderately soluble	
19	L128643	1	3	0	29.54	5.39	5.02	−6.39	Poorly soluble	
20	R461083	2	3	2	59.59	3.93	3.82	−5.59	Moderately soluble	

The BOILED-Egg is a simple yet intuitive model for predicting small molecules’ oral bioavailability (Daina & Zoete, 2016). When we plotted WLOGP and TPSA of the virtual screening hits on the BOILED-Egg (Fig. 2), 11 ligands were inside the egg, the area representing suitable physicochemical space for oral bioavailability. In the context of COVID-19 treatment, candidate compounds in the egg white, which implies human intestinal absorption (HIA) without blood-brain barrier (BBB) permeation, would be preferred for quicker drug development. Four molecules inside the yellow are predicted to be distributed in the brain tissue. Nonetheless, these four compounds seem to be P-glycoprotein (PGP) substrates, and thus, likely to be effluated from the central nervous system. Although nine molecules are in the gray area, they are still close to the egg’s white and would gain better bioavailability profiles during a drug development phase. Together, most of the hits from the MyriaScreen Diversity Library II virtual screening possess optimum physicochemical characteristics for oral bioavailability.

Figure 2 TPSA and WLOGP of top 20 ligands plotted on the BOILED-Egg.

Five major isoforms of cytochromes P450 (CYP1A2, CYP2C19, CYP2C9, CYP2D6, CYP3A4) profoundly impact drug metabolism and elimination. Consequently, these isozymes are key regulators of drug–drug interactions which in turn can dictate efficacy and adverse effects. Table 3 provides data on whether top virtual screening hits can inhibit key CYP isozymes. We found that molecules #17 and #20 are likely to exhibit greater drug–drug interactions as they would inhibit four and five isozymes, respectively. On the other hand, #9 and #12 are inhibitors for none of these metabolic enzymes. Table 3 also shows predicted plasma half-life (T1/2) and clearance of the short-listed molecules.

Table 3 Predicted metabolic and elimination profiles of top ligands.

		Inhibitor	Elimination	
Rank	Ligand ID	CYP1A2	CYP2C19	CYP2C9	CYP2D6	CYP3A4	T1/2 (h)	Clearance (ml/min/kg)	
1	R897698	No	Yes	Yes	No	Yes	1.825	0.749	
2	ST031238	No	Yes	Yes	No	No	1.81	0.83	
3	ST042014	No	No	Yes	No	No	1.71	0.44	
4	ST018363	No	Yes	Yes	No	Yes	1.83	1.01	
5	L363340	Yes	Yes	Yes	No	No	1.7	1.48	
6	ST031351	No	Yes	Yes	No	No	1.61	0.8	
7	L220477	No	Yes	Yes	No	Yes	2.05	1.53	
8	R679445	No	Yes	Yes	No	No	2.03	0.91	
9	ST000954	No	No	No	No	No	1.78	0.8	
10	R872172	Yes	Yes	Yes	No	No	1.98	1.37	
11	ST074801	No	Yes	Yes	No	Yes	1.94	1.3	
12	ST088323	No	No	No	No	No	0.99	0.75	
13	ST018407	No	Yes	No	No	No	1.87	1.52	
14	S51765	No	No	No	No	Yes	1.94	1.27	
15	ST074799	No	Yes	Yes	No	Yes	2.07	1.35	
16	R818984	Yes	Yes	No	No	No	2.21	1.25	
17	ST094780	No	Yes	Yes	Yes	Yes	1.65	1.16	
18	ST020475	Yes	Yes	Yes	No	No	1.37	0.78	
19	L128643	Yes	Yes	No	No	No	2.11	1.44	
20	R461083	Yes	Yes	Yes	Yes	Yes	2.06	1.82	

Lipinski’s rule of five (Lipinski et al., 2001) is extensively used in predicting druglikeness of small molecules. A better plasma membrane permeability is assumed when a compound obeys the following criteria: MW≤ 500, MLOGP ≤, N or O ≤ 10, and NH or OH ≤ 5. As expected, all of the top 20 ligands followed the Lipinski’s rule (Table 4). Most compounds also agreed with other models of druglikeness, namely Ghose, Viswanadhan & Wendoloski (1999), Veber et al. (2002), Egan, Merz & Baldwin (2000), and Muegge, Heald & Brittelli (2001).

Table 4 Drug likeness of top ligands.

Rank	Ligand ID	Lipinski	Ghose	Veber	Egan	Muegge	
1	R897698	Yes	No	Yes	Yes	No	
2	ST031238	Yes	Yes	No	No	Yes	
3	ST042014	Yes	No	Yes	No	Yes	
4	ST018363	Yes	No	Yes	No	No	
5	L363340	Yes	No	Yes	Yes	No	
6	ST031351	Yes	Yes	Yes	No	Yes	
7	L220477	Yes	No	Yes	Yes	Yes	
8	R679445	Yes	No	No	No	No	
9	ST000954	Yes	No	No	No	Yes	
10	R872172	Yes	No	Yes	No	No	
11	ST074801	Yes	No	Yes	Yes	Yes	
12	ST088323	Yes	Yes	Yes	No	Yes	
13	ST018407	Yes	No	Yes	No	No	
14	S51765	Yes	No	Yes	Yes	Yes	
15	ST074799	Yes	No	Yes	Yes	Yes	
16	R818984	Yes	No	Yes	No	No	
17	ST094780	Yes	No	Yes	Yes	Yes	
18	ST020475	Yes	Yes	Yes	Yes	Yes	
19	L128643	Yes	Yes	Yes	Yes	No	
20	R461083	Yes	Yes	Yes	Yes	Yes	

We next computed toxicity profiles of the ligands using the ADMETlab webserver (Dong et al., 2018b), and OSIRIS Property Explorer (Sander, 2017). Table 5 demonstrates that nine of the top ligands could show high toxicities. To note, #1 molecule (R897698) is predicted to have medium cardiac and mutagenic toxicities and high tumorigenicity. On the other hand, #14 compound (S51765) seems to be a safer lead without any major toxicity.

Table 5 Toxicity profiles of top ligands.

		ADMETlab	OSIRIS	
Rank	Ligand ID	hERG blocker	Hepatotoxicity	Ames mutagenicity	Mutagenesis	Tumorigenesis	Irritant	Reproductive effect	
1	R897698	Medium	Low	No	Medium	High	No	No	
2	ST031238	Low	Low	High	No	No	No	No	
3	ST042014	Medium	Medium	No	High	High	No	No	
4	ST018363	Low	Low	No	No	No	Medium	No	
5	L363340	Medium	Low	No	No	No	No	No	
6	ST031351	Low	Medium	High	No	No	No	No	
7	L220477	Medium	Medium	No	No	No	No	Medium	
8	R679445	Medium	Low	No	No	No	No	No	
9	ST000954	Low	No	No	No	No	No	High	
10	R872172	Medium	Low	Low	No	No	No	No	
11	ST074801	Medium	Low	No	No	No	No	No	
12	ST088323	Low	High	Low	High	High	No	Medium	
13	ST018407	Medium	Medium	No	No	No	Medium	No	
14	S51765	No	No	No	No	No	No	No	
15	ST074799	Medium	Low	No	No	No	No	Medium	
16	R818984	Medium	Low	Low	No	High	No	No	
17	ST094780	Medium	Low	No	No	No	No	No	
18	ST020475	Medium	High	No	High	High	No	No	
19	L128643	Medium	Low	Low	High	High	No	No	
20	R461083	Medium	Medium	No	No	No	No	No	

Protein-ligand interaction analysis

When we considered AMDE/Tox profiles of the top 20 hits from the virtual screening, four compounds stand out: L220477, R872172, ST074801, and S51765 (Fig. 3). These molecules have physicochemical properties suitable for oral bioavailability, are predicted not to cross the BBB, and seem to pose lower toxicity risks. Molecular docking confirmed that these four ligands can occupy the active sites of the SARS-CoV-2 main protease (Fig. 3A). Figure 3B shows multiple interactions of 3CL-PRO with the inhibitor 11a in 6LZE. L220477, R872172, ST074801, and S51765 also interact with the critical residues of the protease. Radar charts depict that lipophilicity, size, polarity, insolubility, insaturation, and flexibility of these compounds favor gastrointestinal absorption (Figs. 3D, 3G, 3J and 3M). Interestingly, S51765 resides entirely in the physicochemical space for oral bioavailability (Figs. 3L and 3M). It is also tempting to note that this molecule exhibits the least toxicity risks among the top 20 hits (Table 5).

Figure 3 Docking conformations, physicochemical properties, and protein-ligand interactions for the best four molecules.

(A) Best docking poses of the ligands from virtual screening. In 6LZE, 11a is the co-crystallized ligand. (B) Interactions of 3CL-PRO and the ligand 11a in 6LZE. (C–N) Structure, physicochemical properties, and protein-ligand interactions of L220477 (C and D), R872172 (F–H), L220477 (I–K), and S51765 (L–N). The colored zone in radar charts (D, G, J, and M) indicates suitable physicochemical space for oral bioavailability. LIPO, lipophilicity (XLOGP3); SIZE, molecular weight (g/mol); POLAR, polarity (TPSA); INSOLU, insolubility (LogS); INSATU, insaturation (fraction Csp3); FLEX, flexibility (number of rotatable bonds).

Molecular docking with AutoDock4.2

Although both AutoDock Vina and AutoDock4.2 are widely used for molecular docking and outperform many docking tools in scoring performance, there is a speed-accuracy trade off (Durrant et al., 2013; Wang et al., 2016; Gaillard, 2018; Nguyen et al., 2020). Compared to Vina, AutoDock4.2 was found to generate superior binding affinity (Nguyen et al., 2020). We performed flexible docking for the reference ligand 11a and the best four molecules from our virtual screening. The most negative binding energy was obtained for 11a (−11.23 kcal/mol) followed by L220477 (−10.39 kcal/mol), R872172 (−10.26 kcal/mol), ST074801 (−10.17 kcal/mol), and S51765 (−10.06 kcal/mol). Computed inhibition constants were 5.85, 24.03, 30.11, 35.29, and 42.55 nM for 11a, L220477, R872172, ST074801, and S51765, respectively. These results indicated the best four compounds from our virtual screening were almost identical in terms of AutoDock4.2 binding affinity.

Validation of protein-ligand binding with molecular dynamics simulations

MD simulation studies have significant positive impacts on the drug discovery process (Ganesan, Coote & Barakat, 2017; Liu et al., 2018; Guterres & Im, 2020). We performed duplicated 50-ns MD simulations for AutoDock4.2-generated protein-ligand complexes to validate interactions of the candidate molecules with the SARS-CoV-2 main protease. As a reference, we included the crystal structure 6LZE, where the main protease is complexed with the ligand 11a. The solvent and ions of the simulation systems converged to a minimum energy level within 1,500 minimization steps and subsequently attained NVT and NPT equilibria (Fig. S2). We analyzed the simulation trajectories to predict spatial fluctuations of the protein and ligands in complexes, and results are summarized in Fig. 4 and Fig. S3, for the first and the second simulation, respectively. In the first simulation, mean RMSD values (± SD) of the 3CL-PRO’s C-α atoms were 2.19 (± 0.64), 1.92 (± 0.27), 2.1 (± 0.36), 2.92 (± 1), and 1.69 (± 0.23) angstroms for complexes with 11a, L220477, R872172, ST074801, and S51765, respectively (Fig. 4A).

Figure 4 Spatial fluctuations of protein and ligands during molecular dynamics simulations of complexes.

(A) C-alpha RMSD (running averages) for 3CL-PRO in complexes. (B) Ligand RMSD (running averages) in complexes. (C) C-alpha RMSF for 3CL-PRO in complexes. (D) Ligand RMSF in complexes.

In the first simulation, the protein in the 3CL-PRO-11a complex showed initial fluctuations and the reference ligand 11a remained close to the binding pocket after an initial displacement (Figs. 4A and 4B). Although the protein was fairly stable with R872172 and L220477 (Fig. 4A), Ligand RMSD values indicate wide fluctuations of the compounds (Fig. 4B). When complexed with ST074801, 3CL-PRO seemed to become very unstable at the end of the simulation and the ligand exhibited substantial fluctuations, indicating overall instability of the complex. On the other hand, the 3CL-PRO-S51765 complex showed considerable stability (Fig. 4B).

Mean (± SD) RMSF values of alpha carbon atoms were 1.4 (± 0.57), 1.13 (± 0.57), 1.16 (± 0.65), 1.67 (± 0.89), and 0.96 (± 0.54) angstroms for 11a, L220477, R872172, ST074801, and S51765, respectively (Fig. 4C). The RMSD of all four candidate molecules from the protein backbone were very low, even lower than that of the reference ligand (Fig. 4B). We did not observe any apparent differences in the ligand RMSF (Fig. 4D).

The reference ligand showed a higher RMSD in the second simulation. Compared to the first simulation (Fig. 4B), S51765 also exhibited a higher RMSD value in the second simulation (Fig. S3B). When we repeated the simulation three more times, S51765 indicated considerable stability of the complex with low RMSD values (Fig. S4B). 3CL-PRO became unstable with R72172 and the ligand left the cavity (Figs. S3A and S3B). Interestingly, L220477 showed the least fluctuations (Fig. S3B). However, this ligand moved out of the binding pocket in repeated MD simulations (Figs. S4E and S4F). ST074801 also could not form a stable complex (Figs. S3B, S4C and S4D).

To have a closer look at the binding modes, we extracted frames at every 10 ns from the trajectories and rendered the ligands at the binding cavity (Fig. 5). Interestingly, the reference ligand 11a showed initial displacement at the binding cavity from 0 ns to 10 ns while maintaining contacts with HIS41 throughout the simulation. At around 40 ns, 11a seemed to momentarily move away from GLU166. L220477 (Figs. 5B1–5B6) and R872172 (Figs. 5C1 and 5C6) showed erratic fluctuations indicating unstable complex formation. The conformation of ST074801 in the binding cavity changed significantly during the first 10 ns of simulation (Figs. 5D1–5D6). Intriguingly, the molecule S51765 settled very well in the cavity following a slight displacement at the beginning (Figs. 5E1–5E6). We further analyzed binding poses of S51765 in duplicate simulations (Fig. S5). In one case (simulation-2, Figs. S5A1–S5A6), the binding poses differed from other simulation. Nevertheless, S51765 showed consistency in most of the MD simulations, suggesting stability of the 3CL-PRO-S51765 complex.

Figure 5 Protein-ligand binding modes in MD simulations of best ligands.

Protein-ligand conformations at every 10 ns of simulation for 11a (A1–A6), L220477 (B1–B6), R872172 (C1–C6), ST074801 (D1–D6), and S517656 (E1–E6).

We next analyzed the hydrogen bonds between 3CL-PRO and the selected ligands setting 3 Å as the maximum donor-acceptor distance in VMD. Numbers of hydrogen bonds were plotted over the simulation period in Figs. 6A–6E (first simulation) and Figs. S6A–S6E (second simulation). Occupancy of hydrogen bonds were shown in Table S2. We also plotted hydrogen bond occupancy by ligands (Fig. 6F; Fig. S6F) and by major amino acids in the binding pocket of 3CL-PRO (Fig. 6G; Fig. S6G). Clearly, the reference ligand 11a (Fig. 6A) exhibited the highest interactions over time, which was followed by S51765 (Fig. 6E) and ST074801 (Fig. 6D). Seemingly, L220477 and R872172 failed to establish sufficient hydrogen bonding for making stable complexes (Figs. 6B and 6C). Detail calculations identified residues HIS41, GLU166, and CYS145 as the best hydrogen bond donors for the reference ligand 11a in the crystal structure (Fig. 6G; Table S2). S51765 exhibited the highest occupancy for GLU166 followed by GLN189, MET165, and CYS145 whereas, ST074801 showed the highest interactions with GLN189 (Fig. 6G; Table S2).

Figure 6 Analysis of hydrogen bonding interactions for best ligands.

(A–E) Number of hydrogen bonds between the ligand and 3CL-PRO during the simulation period. (F) Occupancy of hydrogen bonding for the best ligands. (G) Occupancy of hydrogen bonding of the ligand with some important residues at the active site of 3CL-PRO.

We computed distances between the donor-acceptor atoms for the hydrogen bonds with the highest occupancy using the distance module of Gromacs (Table 6). The distance was below 3 Å only in the 3CL-PRO-S51765 complex (Table 6). Intriguingly, the distance was highly consistent for this complex over the entire simulation period (Fig. 7E), whereas, the distance showed a high degree of fluctuation for other complexes (Figs. 7A–7D). We also calculated the highest occupancy protein-ligand hydrogen-bond distances for duplicate simulations of the 3CL-PRO-S51765 complex (Fig. S7). Except for simulation-2, the computed distances were highly indicative of a stable complex formation.

Table 6 Distances between the ligand and the key amino acid residues forming high-occupancy hydrogen bonds.

Ligand	Donor	Acceptor	Occupancy (%)	Average distance (nm)	Standard deviation (nm)	
11a	HIS41	11a	12.15	0.3494	0.10415	
L220477	L220477	ASP187	5.18	0.35085	0.0853	
R872172	THR24	R872172	2.19	0.62455	0.34955	
ST074801	GLN189	ST074801	4.18	0.39249	0.08995	
S51765	GLU166	S51765	55.38	0.27349	0.01284	
S51765 (Simulation-2)	GLN189	S51765	7.77	0.4569	0.14613	
S51765 (Simulation-3)	GLU166	S51765	57.97	0.29025	0.0559	
S51765 (Simulation-4)	GLU166	S51765	37.65	0.3124	0.06403	
S51765 (Simulation-5)	GLU166	S51765	49.00	0.29039	0.02189	

Figure 7 Key distances (running averages of 20 ps) between the ligand and the key amino acid residues of the target protein.

Distances (in angstrom) are plotted against time for (A) 11a and HIS41, (B) L220477 and ASP187, (C) R872172, (D) ST074801, and (E) S51765.

To validate protein-ligand interactions further, we extracted two important energy terms from the GROMACS MD simulation trajectories: short-range Coulomb (Coul-SR) and short-range Lennard–Jones (LJ-SR) (Fig. 8). All ligands had negative values for both of the energies. Over the 50-ns simulation period, the means (± SD) of the sum of Coul-SR and LJ-SR were −200 (± 28), −162 (± 24), −169 (± 24), −250 (± 23), and −194 (± 20) kJ/mol for 11a, L220477, R872172, ST074801, and S51765, respectively (Fig. 8A). These results indicated that S51765 is capable of forming a thermodynamically stable complex with the SARS-CoV-2 3CL-PRO.

Figure 8 Protein-ligand interaction energies from molecular dynamics simulations for complexes of best ligands.

(A) Average short-range Coulomb (Coul-SR) and short-range Lennard–Jones (LJ-SR) energies for the complexes. Error bars show standard deviations. (B–F) Coul-SR and LJ-SR for the complexes over the simulation period.

MMPBSA binding energy calculation

Binding free energy is a reliable measure of protein-ligand interactions. The Molecular Mechanics Poisson-Boltzmann Surface Area (MMPBSA) approach efficiently recapitulates the binding capacity of a small molecule to the target (Kumari, Kumar & Lynn, 2014; Wang et al., 2018). We computed the binding energy (kJ/mol) using the g_mmpbsa tool (Kumari, Kumar & Lynn, 2014) and results are presented in Table 7. Total binding energy values were −72.95 and −70.10 kJ/mol for the reference ligand 11a and S51765, respectively. Compared to 11a, S51765 showed slightly higher van der Wall energy (170.17 kJ/mol vs. 159.31 kJ/mol) and slightly lower electrostatic energy (−23.35 kJ/mol vs. 32.73 kJ/mol). There was no apparent difference in the polar solvation energy. Overall, the free energy signature of S51765 was almost identical with that of the reference ligand.

Table 7 Free energy calculations for the best ligand and the reference ligand.

Energy terms	11a:3CL-PRO complex	S51765:3CL-PRO complex	
Simulation period	Simulation period	
20–25 ns	45–50 ns	Mean	20–25 ns	45–50 ns	Mean	
van der Waal energy (kJ/mol)	−164.80	−175.54	−170.17	−152.02	−166.60	−159.31	
Electrostatic energy (kJ/mol)	−22.72	−23.98	−23.35	−30.69	−34.77	−32.73	
Polar solvation energy (kJ/mol)	140.25	140.87	140.56	142.61	139.12	140.87	
SASA energy (kJ/mol)	−19.55	−20.45	−20.00	−19.02	−18.82	−18.92	
Binding energy (kJ/mol)	−66.81	−79.09	−72.95	−59.12	−81.08	−70.10	

Discussion

To leave no stone unturned in discovering cures for COVID-19, the scientific community is deploying diverse approaches, from in silico to in vitro and from in vivo to clinical. We virtual screened the MyriaScreen Diversity Library II, an unexplored chemical space in the fight against the deadly SARS-CoV-2. This rich compound library from Sigma–Aldrich harbors 10,000 druglike entities encompassing diverse chemotypes (Hole et al., 2015; Njikan et al., 2018; Prado et al., 2018; Jain et al., 2020). A comprehensive in silico approach helped us identify at least four novel leads to design antiviral agents for treating COVID-19. Our computational study will accelerate future in vitro and in vivo experiments to discover antiviral agents for COVID-19.

Target-based virtual screening studies often rely on a single crystallographic structure. With the rapidly evolving COVID-19 situation, we see a surge in crystallographic studies of viral proteins. Now one has the luxury to choose from more than two hundred X-ray crystallographic structures available in the Protein Data Bank (www.rcsb.org) for the replicase polyprotein 1ab (also known as pp1ab) (UniProt accession code P0DTD1), the precursor of SARS-CoV-2 3CL-PRO. Inhibitor-bound crystal structures provide substantial insight into the protein’s active sites to devise target-based inhibitors. Nevertheless, an X-ray crystallographic structure is a snapshot of a particular state, whereas the protein is very much dynamic and can adopt numerous forms in vivo. Conformational changes often occur from the unbound (also known as apo) to the substrate-bound (also known as holo) state. Moreover, a protein can undergo structural alterations depending on intra- and inter-molecular interactions.

Presumably, virtual screening of thousands of compounds using only a single target structure is very prone to miss potential ligands. Instead, we used 33 conformations of the 3CL-PRO from molecular dynamics simulations of three high-resolution crystallographic structures (PDB IDs 6LZE, 6M0K, and 6YB&). We feel that this attempt was rewarded. Ten hits from the combined screening were absent in individual top lists for 6LZE and 6M0K (Table S3). Again, we would have missed 15 of the combined top-ranked molecules if we would consider only 6YB7. The ligand S51765 ranked 135, 88, and 22 when only 6LZE, 6M0K, and 6YB7, respectively, were used singly. Intriguingly, this very ligand turned out to one of the best potential leads in this study. Seemingly, employing many biologically relevant structures of the same target protein can enable capturing potential ligands that would otherwise remain unidentified.

Computational ADMET prediction can profoundly accelerate drug discovery programs by eliminating compounds with unfavorable physicochemical characteristics and toxicity profiles at an earlier stage. In our study, we used SwissADME (Daina, Michielin & Zoete, 2017), ADMETlab (Dong et al., 2018b), and OSIRIS (Sander, 2017), which are some of the most advanced and widely used tools (Ferreira & Andricopulo, 2019; Kar & Leszczynski, 2020). However, validation of our ADMET prediction would be difficult as there is no recognized 3CL-PRO inhibitors with known clinical data.

Intriguingly, the molecule S51765 is a macrocycle with 19 atoms in the ring. A recent study also identified a macrocyclic biomolecule (PubChem ID: 118098670) as a putative 3CL-PRO inhibitor through screening of protease inhibitors (Havranek & Islam, 2020). Another macrocyclic protease inhibitor Danoprevir (DrugBank accession number: DB11779), an antiviral agent, was used in a clinical trial for COVID-19 (ClinicalTrials.gov Identifier: NCT04345276). Macrocycles present both an opportunity and a challenge for computational drug discovery. This group of compounds are emerging as promising leads which offer high bioavailability with enhanced affinity and selectivity for drug targets (Driggers et al., 2008; Mallinson & Collins, 2012; Heinis, 2014). Although large cyclic compounds are generally difficult to model using docking tools, their active conformations could be obtained with higher confidence when molecular dynamics-based computation methods are employed (Sindhikara et al., 2017; Ugur et al., 2019).

Since the outbreak of the COVID-19 outbreak, many computational studies have been conducted to unveil potentials 3CL-PRO inhibitors from diverse sources including FDA-approved drugs, natural products, synthetic small molecules, and synthetic peptides. For example, in silico screening identified novel inhibitors from flavonoids (Gorla et al., 2020; Batool et al., 2020), marine products (Gentile et al., 2020), protease inhibitors (Havranek & Islam, 2020; Keretsu, Bhujbal & Cho, 2020), and commercial chemical libraries (Gimeno et al., 2020; Ibrahim et al., 2020; Uniyal et al., 2020). To our knowledge, no other study screened the MyriaScreen Diversity Library II for 3CL-PRO.

Virtual screening through molecular docking has several limitations including variability in predicted scores (Corbeil, Williams & Labute, 2012; Koes, Baumgartner & Camacho, 2013). To circumvent the caveats partially, we adopted a number of measures. We attempted to minimize false positives by comparing active-decoys, using multiple target structures, and repeating molecular docking. We next enriched the top ligands by careful ADMET profiling. Finally, we analyzed protein-ligand interactions through duplicated MD simulations and free energy calculations. Conceivably, our in silico study could be an adjunct to, not a substitute for, experimental validation of inhibitors for SARS-CoV-2 3CL-PRO.

Conclusions

The COVID-19 pandemic makes it imperative to find safe and effective remedies at the earliest possible time. Computational studies can accelerate antiviral drug discovery by screening huge small molecule libraries and providing leads for further development. In this study, we attempted two goals, exploring a rich chemical library and maximizing the available structural information of the target protein SARS-CoV-2 3CL-PRO. To mimic the dynamics in biological environments, we generated many target conformations through MD simulations of three high-resolution X-ray crystallographic structures of the viral protease. Subsequent virtual screening of 10,000 druglike small molecules in the MyriaScreen Diversity Library II unveils 20 candidate ligands against a total of 33 conformations of 3CL-PRO. We identified four promising leads via scrupulous physicochemical, biopharmaceutic, and toxicity profiling of top-ranked compounds (Tables 1–5). Visual inspection of protein-ligand interactions also suggested that those four molecules could inhibit the SARS-CoV-2 main protease (Fig. 3).

We validated protein binding of the best four molecules by duplicated 50-ns MD simulations (Figs. 4–8). Figure 5E1–E6 clearly shows that S51765 could form a stable complex since the ligand was confined in the binding pocket of 3CL-PRO with only a subtle fluctuation during the simulated period. Hydrogen bonding is the most ubiquitous non-bonded interactions in ligand binding (Böhm & Schneider, 2003; Williams & Ladbury, 2005). Interestingly, S51765 exhibited significant hydrogen bonding interactions (Fig. 7E) involving key residues for inhibitor binding of 3CL-PRO (Zhang et al., 2020; Jin et al., 2020). This was also substantiated by favorable interaction energies for S51765 (Fig. 8F; Table 7). Together, our comprehensive in silico studies present S51765 as a promising candidate molecule for developing 3CL-PRO inhibitors.

Supplemental Information

Supplemental Information 1 Re-docking of 11a to 6LZE with AutoDock Vina (the crystal structure is in marine blue and the docked complex is in bright oragne).

Click here for additional data file.

Supplemental Information 2 Preparation of MD simulation systems for complexes.

(A) Energy minimization. (B) NVT equilibration. (C) NPT equilibration.

Click here for additional data file.

Supplemental Information 3 Analysis of RMSD and RMSF for best ligands in 2nd MD simulations.

Click here for additional data file.

Supplemental Information 4 Fluctuations of protein c-alpha atoms and ligands during the duplicated MD simulations.

c-alpha RMSD and ligand RMSD for S51765 (A, B), ST074801 (C, D), and L220477 (E, F).

Click here for additional data file.

Supplemental Information 5 Protein-ligand conformations at every 10 ns in duplicate MD simulations of S51765:3CL-PRO complex.

Click here for additional data file.

Supplemental Information 6 Analysis of hydrogen bonding for best ligands in 2nd MD simulations.

Click here for additional data file.

Supplemental Information 7 Key distances (running averages of 20 ps) between the donor atoms of S51765 and the acceptor atoms of 3CL-PRO in duplicate MD simulations.

Click here for additional data file.

Supplemental Information 8 Validation of AutoDock Vina virtual screening by active-decoy comparison.

Click here for additional data file.

Supplemental Information 9 Details of hydrogen bonding interactions between 3CL-PRO and selected ligands in MD simulation trajectories.

Click here for additional data file.

Supplemental Information 10 Ranks of ligands in virtual screening using single PDB structures.

Click here for additional data file.

Supplemental Information 11 A compressed file containing best ligands in SDF format and PDB files of complexes for MD simulations.

Click here for additional data file.

Supplemental Information 12 MyriaScreen SD File (confidential).

Click here for additional data file.

We are grateful to Dr. Muhammad Maqsud Hossain, Director, NSU Genome Research Institute (NGRI), North South University, Bangladesh for allowing us remote access to the high-performance computing facility of the NGRI.

Additional Information and Declarations

Competing Interests

Author Contributions

Data Availability

The authors declare that they have no competing interests.

Asim Kumar Bepari conceived and designed the experiments, performed the experiments, analyzed the data, prepared figures and/or tables, authored or reviewed drafts of the paper, and approved the final draft.

Hasan Mahmud Reza conceived and designed the experiments, authored or reviewed drafts of the paper, and approved the final draft.

The following information was supplied regarding data availability:

The raw data are available in the Supplemental File. The MyriaScreen Diversity Library II was supplied for review only. It cannot be shared publicly because it is a proprietary database but it can be obtained free of charge by request online at https://www.sigmaaldrich.com/chemistry/chemistry-services/high-throughput-screening/screening-request.html.

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
