# Peer review of "Identification of a novel inhibitor of SARS-CoV-2 3CL-PRO through virtual screening and molecular dynamics simulation"

_PeerJ, doi:10.7717/peerj.11261_

## Round 0.1 · original submission · Major Revisions

We have received three very detailed reviews of your work. Please address all of their points thoroughly. I must also request you to:

A) further describe the stability of the binding modes (e.g. by submitting as Supporting Information figures or coordinates of the complexes at 25 ns intervals, or providing graphs showing the evolution of key protein-ligand distances).

B) The high RMSD of the trace ST074801 in figure 4A suggests that the ligand has left the cavity (or that the overall structure has become very unstable). In either case, this must be discussed and discounted.

C) I am also not sure of what you intend to show in Figure 4B, since ligand RMSDs can be computed in several ways: one can take the ligand structure in the reference frame and see how the ligand conformations along the simulation compare (and in this instance, one is strictly measuring the flexibility of the ligand along the simulation, regardless of whether or not it remains bound), or superpose the protein structure at a given point over the structure at the reference frame and compute the distance in ligand coordinates (and in this case the RMSD contains information about the ligand conformation and ligand position). Please clarify this.

Reviewer 1 ·

Basic reporting

The article has some limitations in terms of language that would need to be improved. The overall structure organization is adequate, but the contents are not fully adjusted to this organization. For example, some studies are not described in the methodology section, but appear only described in the results or in the discussion section (MDs with the complexes).

In general, figures are very clear and informative.

Experimental design

The experimental design involves 100 ns molecular dynamics simulations, from which 10 structures were taken. These were later used for docking with of a collection of 286 small molecules from the MyraScreen Diversity Library II, following an initial virtual screening of the 10,000 molecules of this library against the three X-ray structures of the protein. In silico AMDE/TOX profiling was performed with Swiss ADME and OSIRIS.

Protein-ligand interaction analysis based on 50-ns MD simulations was performed on the top 4 compounds and with the 11a reference compound (this part is not described in the methodology).

No free energy calculations were performed to validate the docking results. No experimental validation was performed.

Details are missing concerning the docking calculations and the MD simulations.

No validation of the docking protocol or of the VS protocol seem to have been done.

Validity of the findings

The validity of the findings is difficult to evaluate as no experimental evaluation of the activity of the top compounds was performed.

It is not clear if only autodock vina was used, or if the top results were re-evaluated with Autodock 4.2. Nevertheless binding affinity or scoring values obtained with different independent methods should be presented. No free energy calculations on the top complexes was performed, not even through MM-pbsda/MM-gbsa protocols.

Given the variability of docking scores it is unclear how strong the predictions of one single docking program can be, even when performed on multiple structures

No validation of the docking conditions through re-docking of the co-crystallized ligand seems to have been done.

No validation of the descrimination ability of the VS protocol seems to have been done through actives/decoys comparison

Additional comments

Minor Points:

Line 115 - indicate how many were this selected ligands for which docking with AutoDock4.2 was done.

Line 129 - please justify the use of a 300 K tempeature instead of 310K

Initial docking of the 10,000 small molecules was done with Vina. The following docking stage, on the MD conformations, was done with Autodock 4.2 (as suggested on line 115) or with Autodock vina (as suggested on line 181)

Docking - indicate box size and conditions employed

MD simulations - please indicate thermostat, barostat, size of water radius around protein, non-bonded interaction cut-off, use of LINCS/SHAKE, specific set of parameters used to model the protein.

Reviewer 2 ·

Basic reporting

The English is generally clear, professional, and all other criteria for basic report have been met to a sufficient standard.

Experimental design

The computational methodology follows well-established protocols, but further details in the Methods are required. These are detailed, together with other closely-related comments, in my general comments to the authors below.

Validity of the findings

The findings are valid, but should be treated as hypotheses, given the limitations in the computational methods employed in predictions of enzyme-complex binding affinity and stability.

Additional comments

This study involved the use of an automated docking method to identify the top binding ligands from a small-molecule database, MyrisScreen Diversity Library II, to the papain-like protease of SARS-CoV-2. This was subsequently followed by ADMET filtering of the top candidate compounds and MD simulation studies to elucidate the stability of the enzyme-ligand complexes.
The methodology appears to follow well-established protocols (but much more details are required- see below), and the predictions are potentially valuable. With the caveat that the results should be viewed in light of the limitations in the computational methods employed (such as whether Autodock Vina is capable of reliably predicting binding affinity values for novel compounds), the work described in this manuscript is sound, and I recommend publication after minor revisions.

My main critique is that the modelling and simulation sections lack key details:
Further details in the molecular dynamics Materials & Methods section are required to ensure reproducibility of the simulations. Standard details include the temperature and pressure coupling algorithms employed; treatment of electrostatic interactions; van der Waals and electrostatics cut-off ranges; integration time steps (which is mentioned in Results, line 153, but should be placed in Methods); and whether bond constraints were employed, and if so, which algorithms were used.
The system set-up also require further details. In particular, what were the PBC box dimensions? And how many TIP3P water molecules, sodiums, and chlorides were included to solvate the protein-ligand complexes?

For the Autodock Vina section, what was the value of the exhaustiveness parameter used? What were the dimensions of the docking sampling box defined around the protein?

Finally, in terms of the methodology, it is stated that, after Autodock Vina calculations, a further Autodock4.2 rigid docking was performed on the selected ligands. It is not clear why this additional docking step was required, and the authors should explain the rationale behind this approach. Additionally, docking parameters and algorithms (such as genetic algorithm parameters) for this subsequent Autodock4.2 step should be provided.

For each of the receptor-ligand complexes, only a single 50 ns trajectory was obtained. Ideally, multiple independent simulations should be performed to improve the reliability of the predicted stability of the binding poses. While computational resources may not be sufficient to enable multiple simulations to be performed for all receptor-ligand complexes in a timely manner, some comment should be made to address the potential uncertainties of the predictions due to the use and analyses of single trajectories.

Reviewer 3 ·

Basic reporting

This is a well-written, well-organized paper describing the virtual screening of a small chemical library against SARS-CoV-2 Mpro protease, one of the most important COVID-19 drug targets. The VS results were primarily validated by molecular dynamics (MD) simulations and were also subjected to computational ADMET screens.

Experimental design

The experimental design follows a "common" virtual screening workflow. However, the manuscript does not sufficiently justify the rationale of this workflow, nor does it express sufficient criticism. Even in the best situations, virtual screening has a relatively low success rate, but this is not mentioned by the authors. The authors do not sufficiently describe how their MD simulations are likely to improve the validity of their hits. They should cite more literature that assesses the quantitative effectiveness and limitations of their methods. The authors do not critically assess the validity of their ADMET predictions.

To aid replication, the authors should deposit sdf files of their top hits as well as the PDB files for their protein MD docking templates.

Validity of the findings

The authors should be aware of the controversy surrounding purely computational VS studies, especially for COVID. Many researchers are skeptical of the value of such work. In light of the PeerJ publication philosophy, I believe such studies are publishable. The authors need to be more critical of the methods and their own results. The authors need to more explicitly state the limitations of their work and that without experimental validation, many results should be considered speculative.

The paper lacks deeper analysis of the results and the molecules identified. How does this study fit into the context of the many other Mpro VS papers that have been published this year? Are the identified molecules useful as drug hits or for further drug design? Are they similar to other molecules that have been identified by VS or experimental screening? Is the authors' docking against multiple MD structures have a precedent in the literature? The authors need to describe validation of this approach and that it actually improves VS performance.

Specific points:

1. S51765 is a macrocycle that is poorly handled by Autodock Vina and most docking softwares due to its many degrees of freedom. This issue is not addressed in the paper.

2. What is the rationale for use of the MyriaScreen Library? What makes it particularly interesting or useful especially compared to the many other vendor libraries available? Has it been previously screened in other studies?

3. The Autodock Vina predicted binding affinities are well-known to poorly correlate with experimental values. They should be supplemented by more effective computational methods for predicting binding affinities, such as MM-GB/PB-SA or more sophisticated rescoring functions.

4. The authors should demonstrate their approach works by performing the same analysis with positive controls, ligands known to bind Mpro.

---

## Round 0.2 · Minor Revisions

I agree with reviewer #3's concerns regarding the stability of the binding mode of ST074801, and I would personally refrain from claiming that this molecule is a good binder unless (like the reviewer asks) I had a very large number of simulations that showed that the unbinding event in one of your simulations was a "fluke". Apart from this, I still have some requests for clarification that you should address:

A) Fig 4A/B and Fig. S3A/B are labelled as depicting the same measurements (Calpha RMSD/RMSF for each system) but they are extremely different. Please check this and clarify.

B) L220477 appears to have the most stable interaction in Fig. S3B, but elsewhere (lines 323 and 333) its binding mode is described as "erratic" or "missing sufficient H-bond interactions", and this description is confirmed by Figure S4. Please check this and clarify.

C) Fig S4 should be moved to the main text since it is the most vivid data of the stability of the binding poses. I think that a similar image for the duplicate simulations should also be provided.

D) I am afraid that Figure S5 is very difficult to interpret since comparisons with superposed single amino acids are confounded by changes in conformation/orientation of that aminoacid: one can easily envisage situations where the ligand remains completely immobile while the reference amino acid (SER46, GLN189, or GLU166) freely rotates around its Calpha-Cbeta bond: in this case, superposing the rotating amino acid in each frame with its position in the original frame will obviously make us think that the ligand is moving wildly about... Please measure the key distances between the amino acids and the ligand in each frame (I am sure your MD package has some function to do that automatically) and depict that set of data instead.

Reviewer 1 ·

Basic reporting

The authors have improved the manuscript significantly.

Experimental design

With the addition of the missing information and the clarification of points raised in the previous revision, the experimental design is now quite clear and can be properly assessed as adequate.

Validity of the findings

The validity of the findings is now justified with the new calculations presented and with the details given.

Additional comments

The authors have made all the changes requested and have included all the methodological information that was missing. The protocol now becomes more clear and reproducible.

Appropriate justifications were presented to all the points raised.

I recommend acceptance of the present version.

Reviewer 3 ·

Basic reporting

The authors have greatly improved the quality of the manuscript by addressing the concerns of the editor and reviewers.

Experimental design

I am still concerned about the MD simulations with ST074801. One run showed instability while an additional run did not. What can we conclude from that? The authors should do more runs (total of 5 at least) and report on the stability.

Validity of the findings

The authors have added statements about the limitations of virtual screening and predicting drug-binding and drug affinity. This is very important so that non-specialists recognize the proper scope of the work.

---

## Round 0.3 · accepted · Accept

Thank you for addressing the final issues. I am glad to accept your manuscript for publication.